# The Effects of Remimazolam and Inhalational Anesthetics on the Incidence of Postoperative Hyperactive Delirium in Geriatric Patients Undergoing Hip or Femur Surgery Under General Anesthesia: A Retrospective Observational Study

**DOI:** 10.3390/medicina61020336

**Published:** 2025-02-14

**Authors:** Jimin Kim, Sangseok Lee, Byung Hoon Yoo, Yun Hee Lim, In-Jung Jun

**Affiliations:** Department of Anesthesiology and Pain Medicine, Sanggye Paik Hospital, Inje University College of Medicine, Seoul 01757, Republic of Korea; kimjeemin518@naver.com (J.K.); s2248@paik.ac.kr (S.L.); twowind@paik.ac.kr (B.H.Y.); painfree@paik.ac.kr (Y.H.L.)

**Keywords:** postoperative delirium, remimazolam, sevoflurane, hip surgery

## Abstract

*Background and Objectives*: Postoperative delirium (POD) is a transient but significant complication in geriatric patients following hip or femur surgery. POD occurs in 19–65% of patients after hip surgeries, with notable risks associated with augmented morbidity, mortality, and prolonged hospitalization. The perioperative administration of benzodiazepines, particularly midazolam, is associated with an increased incidence of POD. Remimazolam, a novel ultra-short-acting benzodiazepine, has potential benefits, such as hemodynamic stability and ease of reversal, but its effect on POD occurrence remains unclear. *Materials and Methods*: This retrospective study investigated patients who were aged 65 years old and older who underwent hip or femur surgery. Following the application of exclusion criteria, 502 patients were grouped according to whether anesthesia was maintained with remimazolam (R group) or sevoflurane (S group). Data regarding patients’ baseline characteristics, anesthetic details, and postoperative outcomes, including the incidence of POD, were gathered and analyzed. Propensity score matching and logistic regression were conducted to identify factors associated with POD and compare outcomes between the two groups. *Results*: Among the 502 patients, POD was observed in 161 (32%). The POD incidence was not statistically significantly different between the groups (*p* = 1.000). A multivariable logistic regression analysis indicated that remimazolam was not a determinant of POD (*p* = 0.860), whereas being male and polypharmacy were (*p* = 0.022; *p* = 0.047). Initial disparities in age and comorbid conditions between the groups were rectified through matching, demonstrating that remimazolam had a similar POD risk to sevoflurane. *Conclusions*: This study showed that remimazolam did not exacerbate the risk of POD in elderly patients undergoing hip or femur surgery. Remimazolam is a reliable anesthetic option for this vulnerable demographic. Also, this study’s results indicated that polypharmacy and being male are POD risk factors, suggesting that meticulous perioperative medication management may help alleviate the risk of POD.

## 1. Introduction

Postoperative delirium (POD) is a transient cerebral dysfunction that alters consciousness, attention, cognition, and perception after surgery. It is associated with adverse outcomes in surgical patients, including increased morbidity and mortality. Among hip surgery patients, 19–65% experience POD [1]. The causes of POD are multifactorial. Patients undergoing hip or femur surgery are prone to POD, because they tend to be older and be on multiple psychological medications, including benzodiazepine, and have multiple comorbidities.

As the population ages, the number of hip surgeries is increasing, leading to a rise in the incidence of POD. POD is associated with increased morbidity and mortality, prolonged hospital stays, worse functional recovery, and long-term cognitive decline. It is also a significant burden on patients, their families, and medical staff, making its prevention crucial.

POD has predisposing and perioperative factors [2]. One of these factors is the perioperative administration of benzodiazepine, such as midazolam, which suppresses the release of the inhibitory neurotransmitter gamma-aminobutyric acid [3,4]. Remimazolam, a novel ultra-short-acting benzodiazepine, is widely used for sedation and general anesthesia [5]. It promotes hemodynamic stability, reduces respiratory depression, and is easily reversible, making it especially suitable for elderly individuals undergoing hip surgery. However, its risk of causing POD is unclear. This study retrospectively compared the effects of remimazolam and inhalational anesthetics (sevoflurane) on the incidence of POD among elderly patients, who are at a higher risk of POD.

## 2. Materials and Methods

### 2.1. Study Design and Participants

This study was a single-center retrospective study and approved by the Institutional Review Board of our institution (No. 2023-03-011). Due to its retrospective design, the Institutional Review Board waived the informed consent requirement. Patients aged 65 years old and older who underwent hip or femur surgery under general anesthesia between January 2020 and March 2023 were included in the study. Demographic data; comorbidities; and operation, anesthetic, and perioperative consultation records were taken from electronic medical records and analyzed.

Patients under the age of 65 or who received spinal or local anesthesia, underwent total intravenous anesthesia with propofol, were administered a combination of remimazolam and inhalational anesthetics, or underwent reoperations were excluded from the study. Patients were grouped according to whether anesthesia was maintained with remimazolam (R group) or sevoflurane (S group). This study was performed in accordance with the Strengthening the Reporting of Observational Studies in Epidemiology statement.

### 2.2. Anesthesia Management

Patients were given remimazolam or inhalational anesthetics sevoflurane according to the preference of the anesthesiologist. Standard monitoring devices, such as three-lead electrocardiograms, noninvasive blood pressure devices, and pulse oximeters, were used for hemodynamic management. Invasive blood pressure monitoring using a radial artery catheter was employed if necessitated by the patients’ underlying medical conditions. Remimazolam was infused at a rate of 6 mg/kg/h until the patient was unconscious, and 0.5–2 mg/kg/h was infused to maintain a bispectral index (Aspect Medical Systems, Inc., Natick, MA, USA) of 40–60 for the R group. Propofol bolus was injected in an amount of 1–2 mg/kg to induce anesthesia, and 1–2.5% sevoflurane was administered to maintain a bispectral index of 40–60 for the S group.

All patients received 0.03–0.2 mcg/kg/min remifentanil for intraoperative analgesia and muscle relaxants, fluid therapy, and blood transfusions. Fentanyl was administered 30 min prior to the end of anesthesia at 0.5–1 mcg/kg. Additionally, depending on the patient’s condition, the anesthesiologist administered acetaminophen at 1.5 mg/kg up to 1 g. Surgery was performed by the same experienced surgeon. At the end of the surgery, the patient’s neuromuscular blockade was reversed using sugammadex, and the endotracheal tube was extubated. Patients in the R group who did not open their eyes 15 min after the discontinuation of remimazolam administration were administered flumazenil (0.2 mg).

Postoperative analgesia was maintained by repeated fentanyl 0.5–1 mcg/kg administration if the patient’s numerical rating score was more than 5 in the postanesthesia care unit. In the ward, patients were given patient-controlled analgesia with a mixture of nefopam hydrochloride 80 mg and ramosetron 0.6 mg with a basal rate of 1 mL/h, bolus 1 mL, and 15 min lock-out time.

### 2.3. Variable Definitions

This study’s variables were the patient characteristics, namely age, gender, height, weight, and alcohol consumption status; comorbidities, namely diabetes mellitus, hypertension, coronary artery disease, asthma, chronic obstructive lung disease, history of cerebrovascular attack, pneumonia, renal failure, and hepatitis; preoperative laboratory findings, namely hemoglobin, albumin, creatinine, sodium, aspartate transaminase, alanine transaminase, and C-reactive protein; Charlson comorbidity index; polypharmacy; and COVID-19 virus infection status (Table 1). Variables related to surgery and anesthesia were the operation type, namely total hip arthroplasty, bipolar hemiarthroplasty, open reduction and internal fixation, closed reduction and internal fixation, or incision and drainage; surgical emergency status; anesthesia duration; operation duration; amount of fluid infused; amount of blood transfused; urine output; estimated blood loss; intraoperative hypotension, defined based on the use of the vasoconstrictor phenylephrine or ɑß-agonist ephedrine; and preoperative and postoperative pain scores (Table 2). Postoperative data were the POD incidence, hospital stay duration, intensive care unit stay duration, and 30-day mortality (Table 3).

Alcohol consumption status was defined as having regularly consumed alcohol before hospitalization. Diabetes mellitus was defined as having a history of uncontrolled blood glucose levels, with a preoperative history of taking anti-hyperglycemic medications. Hypertension was defined by a diagnosis of hypertension and a preoperative history of taking antihypertensive medication. Coronary artery disease was defined as an ischemic heart disease diagnosis by a cardiologist. Asthma and chronic obstructive lung disease were defined based on diagnosis by spirometry and a clinical symptom. Cerebrovascular attack was defined by a history of stroke. Pneumonia was defined by radiological examination results that were consistent with pneumonia. Renal failure was defined by chronic renal failure diagnosis based on the estimated glomerular filtration rate. Hepatitis was defined as a blood test indicating the presence of hepatitis B or C virus. The Charlson comorbidity index was calculated using age, myocardial infarction, congestive heart failure, peripheral vascular disease, cerebrovascular attack, dementia, chronic pulmonary disease, connective tissue disease, peptic ulcer, liver disease, diabetes mellitus, hemiplegia, chronic kidney disease, leukemia, lymphoma, and acquired immunodeficiency syndrome, each of which was weighted according to its potential influence on mortality. Polypharmacy was defined as the concurrent use of several prescription medications, usually five or more taken daily. COVID-19 infection was defined as COVID-19 virus infection diagnosis within the six weeks before surgery. Preoperative pain was measured one day before the operation, and postoperative pain was measured on arrival in the postanesthesia care unit after the operation.

### 2.4. POD Assessment

The primary outcome was POD within five days after surgery. Attending physicians who determined that patients were experiencing POD had consulted with a psychiatrist. The psychiatrist diagnosed POD using the *Diagnostic and Statistical Manual of Mental Disorders, Fifth Edition*. Preoperative delirium was assessed by the attending physician for the period between hospital admission and the start of surgery. We reviewed electronic medical records to determine whether POD occurred.

### 2.5. Statistical Analysis

Statistical evaluations were performed utilizing R version 4.11 (R Foundation for Statistical Computing, Vienna, Austria). Data distribution normality was assessed by Kolmogorov–Smirnov tests. Demographic characteristics, perioperative data, and postoperative outcomes, including POD incidence, were compared between the groups using Student’s t-tests for continuous variables and chi-squared tests for categorical variables. Mann–Whitney U or Fisher’s exact test was used to analyze nonparametric data as appropriate. Categorical variables are reported as counts and percentages, while continuous variables are reported as mean ± standard deviation or median (interquartile range).

Each variable presented in Table 1 was independently subjected to logistic univariate analysis to identify POD risk factors. Variables with a *p*-value < 0.1 were incorporated into the multivariable analyses, and their odds ratios were computed. Propensity score matching was executed to mitigate the effects of potential confounding variables caused by systematic disparities between the two groups. A *p*-value < 0.05 was regarded as statistically significant for all analyzed data.

## 3. Results

A total of 833 patients undergoing hip surgery from January 2020 to March 2023 were recruited for this study, of which 331 were excluded because 216 patients were younger than 65, 67 received an alternative anesthesia method, 6 were administered two types of anesthetics simultaneously, and 42 underwent reoperation. After exclusion, 502 patients were assessed for eligibility for analysis. After propensity score matching, 93 patients were included in the R group and 93 in the S group (Figure 1).

Table 1 presents the patient baseline characteristics, and Table 2 presents data related to surgery and anesthesia before and after matching. Before adjustments, the R group had a mean age that was 3.3 years older and a higher incidence of comorbidities and was more likely to receive blood transfusions than the S group. After matching, there were no significant differences in patient baseline characteristics. Table 3 presents the postoperative outcomes. Of the 502 patients who were analyzed in this study, POD was observed in 161 (32%). The POD incidence was not significantly different between the two groups before (R group: 66/194 patients [33.7%]; S group: 95/308 [30.7%]; *p* = 0.475) or after matching (*p* = 1.000) (Table 3).

Table 4 shows the multivariable logistic regression analysis results for POD incidence. The univariate logistic regression analysis showed that the POD incidence was positively correlated with being male, polypharmacy, hypertension, hemoglobin, albumin, and C-reactive protein (*p*-value < 0.1). The multivariate logistic regression analysis showed that remimazolam administration was not correlated with POD incidence. Being female was negatively correlated with POD incidence (*p* = 0.022; OR = 0.337; 95% CI: 0.133–0.853), and polypharmacy was statistically significantly positively correlated with POD incidence (*p* = 0.047; OR = 1.092; 95% CI: 1.001–1.191) (Table 4).

## 4. Discussion

This study’s results suggest that remimazolam was not more likely to cause POD in geriatric patients undergoing hip surgery than sevoflurane. The literature indicates that benzodiazepines may increase the risk of POD, but this study’s results are consistent with emerging studies that indicate that remimazolam is a safe alternative anesthetics for elderly surgical candidates.

Several studies have indicated that remimazolam is not more likely to cause POD than other anesthetics. Fujimoto et al. (2024) conducted a retrospective review and found that remimazolam was less likely to cause POD than propofol, sevoflurane, and desflurane in patients undergoing femur fracture surgery [6]. Yang et al. (2023) also reported fewer incidences of POD among patients who received remimazolam than among those who were administered propofol for orthopedic surgery [7]. Two cardiovascular surgery studies reported fewer incidences of POD among those who were administered remimazolam than among those who were administered other anesthetics [8,9].

It is not yet entirely clear why remimazolam does not raise the risk of POD while other benzodiazepines do. One possible explanation is its unique metabolic characteristic of being broken down by nonspecific tissue esterases, giving it ultra-short-acting characteristics and body wash-out times. Previous studies have shown that benzodiazepines with various half-lives have different effects on the incidence of POD [10,11]. While long-acting benzodiazepines increased the risk of delirium, short-acting benzodiazepines did not. Remimazolam has an ultra-short half-life of 0.5–2 min and an elimination half-life of 37–53 min, reducing its side effects, such as POD [12]. Another possible explanation is remimazolam’s inflammation-attenuating effects and ability to maintain hemodynamic stability, both of which reduce stress and improve postoperative outcomes. Surgery upregulates pro-inflammatory cytokines, which may affect cognitive function, but remimazolam is reported to decrease inflammatory mediators, such as interleukin-6, tumor necrosis factor-α, and interleukin-1β [13]. Sun et al. (2023) reported that adding low-dose remimazolam attenuated the inflammatory responses of interleukin-6 and tumor necrosis factor-α and produced better pain and stress scores [14]. Further studies are needed to confirm whether remimazolam attenuates inflammation and stress and prevents POD.

Remimazolam offers significant advantages in elderly patients. It is hemodynamically stable, causing fewer incidences of hypotension in elderly patients during anesthesia induction than propofol [15]. As a result, it helps maintain stable cerebral blood flow without severe blood pressure fluctuations, which may reduce the incidence of POD. Moreover, it has rapid onset and offset. These characteristics make it a promising choice for routine use in high-risk surgeries involving elderly patients, such as hip surgeries [16].

As the results of this study show, polypharmacy and being male are risk factors for POD [17]. Most patients who are on multiple medications have multiple comorbidities, and the number of drug interactions is positively correlated with the number of medications, leading to adverse effects, such as a loss of appetite, sedation, and cognitive dysfunction, all of which may contribute to the incidence of POD [18]. Up to 74% of patients are overmedicated, and in this study, 53% of patients were on multiple medications [19]. Although the patients’ comorbidities could have contributed to POD, there was no correlation between Charlson’s comorbidity index and POD incidence in this study, indicating that drug interactions may have been the primary factor behind POD. This result highlights the need to reduce the number of unnecessary medications that elderly patients are on during the perioperative period. There are tools available to help manage polypharmacy and limit inappropriate medication use in elderly patients. Two commonly used tools are the American Geriatrics Society (AGS) Beers Criteria and the Screening Tool of Older People’s Prescriptions (STOPP) and Screening Tool to Alert to Right Treatment (START), collectively known as the STOPP/START criteria [20,21]. The AGS Beers Criteria focus on medications that should be used with caution in elderly patients and provide information on drug–drug interactions. The STOPP/START criteria serve as a guideline for appropriate prescribing to reduce adverse drug events.

Male gender is a known risk factor for POD following surgery. Previous epidemiological studies suggest that male gender may be related to more risk factors affecting cognition-related brain domains, such as obstructive sleep apnea, alcohol dependence, and psychological stress related to illness, all of which may contribute to POD [22]. Additionally, estrogen may play a protective role in individuals with potential cognitive impairment, although the specific mechanisms underlying its effects on POD remain unclear [23].

A peripheral nerve block is undoubtedly important in preventing POD, although we excluded patients who received a peripheral nerve block to focus on the impact of remimazolam on POD. There is consistent evidence that combining a peripheral nerve block with general anesthesia reduces the incidence of POD in elderly patients with hip fractures [24,25]. A meta-analysis of 19 RCTs demonstrated that peripheral nerve blocks significantly lowered the POD incidence (OR: 0.59; 95% CI [0.40–0.87]; *p* = 0.007) [26]. Specifically, the fascia iliaca compartment block (OR: 0.58; 95% CI [0.37–0.91]; *p* = 0.02) and the femoral nerve block (OR: 0.33; 95% CI [0.11–0.99]; *p* = 0.05) were associated with reduced POD risk. Additionally, peripheral nerve blocks significantly reduced pain scores (*p* = 0.002). Pain is a major risk factor for POD, and by utilizing peripheral nerve blocks, elderly patients benefit from better analgesia, reduced inflammatory response, decreased opioid consumption, and, ultimately, a lower incidence of POD. Without a peripheral nerve block, postoperative pain management after general anesthesia is challenging. In future studies, we may explore the combination of peripheral nerve blocks with remimazolam, which we anticipate will result in the lowest POD incidence. If confirmed, this approach could be widely implemented in clinical practice for elderly patients.

This study had two major limitations. The first limitation was its retrospective design, which may have introduced selection bias despite our efforts to control for confounders through propensity score matching. While the matching did balance the patients’ baseline characteristics, certain unmeasured variables such as socioeconomic and nutritional status could still have impacted the results. Hence, as a retrospective single-center study, caution is needed when generalizing the findings more broadly. The second limitation was that POD was identified through observable symptoms and psychiatrist consultations, which may have led to hypoactive POD cases going unreported. Therefore, we are conducting a prospective study using standardized POD assessment tools, such as the Confusion Assessment Method and other supplements, in collaboration with neuropsychiatrists to ensure that all subtypes are detected.

## 5. Conclusions

This study showed that remimazolam may be a safe anesthetic option for geriatric patients undergoing hip surgery, given that it did not increase the incidence of POD compared with sevoflurane. Given its rapid metabolic clearance, hemodynamic stability, and minimal impact on cognitive outcomes, remimazolam holds promise for use in elderly, high-risk surgery patients. Further research, ideally in a prospective, multicenter format, can confirm these findings and elucidate the factors contributing to POD in this patient population.

## Figures and Tables

**Figure 1 medicina-61-00336-f001:**
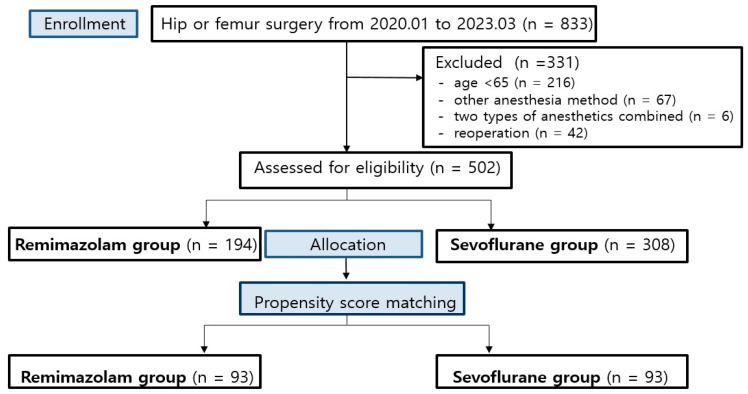
Flow diagram.

**Table 1 medicina-61-00336-t001:** Baseline characteristics.

	Before Matching				After Matching			
Variables	R Group (n = 194)	S Group (n = 308)	*p*	SMD	R Group (n = 93)	S Group (n = 93)	*p*	SMD
Age (years)	83.3 ± 6.5	80.0 ± 7.9	0.0001	0.456	82.3 ± 6.8	81.4 ± 8.0	0.413	0.120
Sex (M/F)	48/146	104/204	0.041	0.199	27/66	31/62	0.635	0.093
Height (cm)	157.2 ± 8.0	157.9 ± 8.8	0.386	0.079	158.2 ± 8.9	157.5 ± 8.6	0.603	0.077
Weight (kg)	56.1 ± 10.9	56.5 ± 10.9	0.676	0.039	57.9 ± 11.5	54.7 ± 10.9	0.052	0.290
Alcohol	14 (7.2)	38 (12.3)	0.092	0.173	10 (10.8)	10 (10.8)	1.000	0.001
Comorbidity								
Diabetes mellitus	79 (40.7)	107 (34.7)	0.209	0.124	32 (34.4)	33 (35.5)	1.000	0.023
Hypertension	152 (78.4)	218 (70.8)	0.076	0.175	71 (76.3)	68 (73.1)	0.736	0.074
Coronary artery disease	3 (1.5)	3 (1.0)	0.681	0.051	1 (1.1)	0 (0)	1.000	0.001
Asthma	9 (4.6)	10 (3.2)	0.578	0.072	3 (3.2)	5 (5.4)	0.721	0.106
Chronic obstructive lung disease	8 (4.1)	5 (1.6)	0.153	0.150	3 (3.2)	4 (4.3)	1.000	0.057
Cerebrovascular attack	24 (12.4)	43 (14)	0.707	0.047	10 (10.8)	11 (11.8)	1.000	0.034
Pneumonia	0 (0)	1 (0.3)	1.000	0.081	0 (0)	0 (0)	1.000	<0.001
Renal failure	37 (19.1)	23 (7.5)	0.0001	0.347	10 (10.8)	11 (11.8)	1.000	0.034
Hepatitis	4 (2.1)	4 (1.3)	0.493	0.059	1 (1.1)	1 (1.1)	1.000	<0.001
Preoperative Lab								
Hemoglobin (g/dL)	11.1 ± 1.5	11.3 ± 1.7	0.233	0.108	11.2 ± 1.6	11.2 ± 1.6	0.897	0.019
Albumin (g/dL)	3.6 ± 0.4	3.4 ± 0.5	0.001	0.315	3.5 ± 0.4	3.5 ± 0.5	0.896	0.020
Creatinine(mg/dL)	0.9 ± 1.0	0.7 ± 0.6	0.016	0.246	0.9 ± 1.0	0.8 ± 0.9	0.765	0.045
Sodium (mEq/L)	136.3 ± 3.1	136.6 ± 2.9	0.297	0.104	136.2 ± 3.1	136.2 ± 3.3	0.958	0.008
Aspartate Transaminase (IU)	27.4 ± 22.3	27.7 ± 12.6	0.867	0.017	26.5 ± 14.0	28.2 ± 14.9	0.434	0.116
Alanine transaminase (IU)	16.4 ± 22.9	16.0 ± 11.6	0.828	0.021	15.9 ± 14.3	14.8 ± 11.5	0.551	0.088
C-reactive protein (mg/dL)	4.0 ± 4.0	3.9 ± 4.5	0.852	0.018	4.0 ± 3.9	3.7 ± 4.3	0.622	0.076
Preoperative delirium	11 (5.7)	12 (3.9)	0.480	0.083	5 (5.4)	6 (6.5)	1.000	0.046
Charlson comorbidity index	5.0 ± 1.2	4.5 ± 1.3	0.0001	0.392	4.8 ± 1.3	4.7 ± 1.5	0.593	0.079
Polypharmacy	8.1 ± 4.6	6.6 ± 4.1	0.002	0.329	8.3 ± 5.1	7.1 ± 4.9	0.166	0.238
COVID-19 infection	20 (10.3)	9 (2.9)	0.001	0.301	4 (4.3)	4 (4.3)	1.000	<0.001

Data are shown as mean ± standard deviation or median (interquartile range) or number (%).

**Table 2 medicina-61-00336-t002:** Data related to surgery and anesthesia.

	Before Matching				After Matching			
Variables	R Group (n = 194)	S Group (n = 308)	*p*	SMD	R Group (n = 93)	S Group (n = 93)	*p*	SMD
Operation type			0.001	0.452			0.988	0.096
Total hip arthroplasty	6 (3.1)	31 (10.1)			4 (4.3)	5 (5.4)		
Bipolar hemiarthroplasty	87 (44.8)	118 (38.3)			35 (37.6)	38 (40.9)		
Open reduction internal fixation	6 (3.1)	11 (3.6)			6 (6.5)	6 (6.5)		
Close reduction internal fixation	91 (46.9)	121 (39.3)			44 (47.3)	40 (43.0)		
Incision and drainage	0 (0)	6 (1.9)			0 (0)	0 (0)		
Other	4 (2.1)	21 (6.8)			4 (4.3)	4 (4.3)		
Emergency operation	3 (1.5)	14 (4.5)	0.120	0.175	3 (3.2)	2 (2.2)	1.000	0.067
Anesthesia duration (min)	130.9 ± 28.9	138.0 ± 40.8	0.024	0.199	134.9 ± 31.9	141.2 ± 42.5	0.256	0.167
Operation duration (min)	69.3 ± 27.1	80.1 ± 36.6	0.0001	0.337	73.1 ± 30.9	79.1 ± 35.9	0.226	0.178
Total infused fluid (mL)	992.5 ±392.2	937.0 ± 468.1	0.153	0.129	1024.5 ± 410.6	1078.2 ±569.1	0.462	0.108
Blood intake (mL)	76.2 ± 128.3	38.4 ± 91.7	0.0001	0.339	76.5 ± 133.6	78.7 ± 122.9	0.905	0.018
Urine output (mL/kg/h)	0.018 ± 0.01	0.019 ± 0.01	0.687	0.038	1.198 ± 1.06	1.17 ± 1.03	0.910	0.017
Estimated blood loss (mL)	178.7 ± 164.2	181.0 ± 188.9	0.882	0.013	177.1 ±147.4	194.1 ± 207.7	0.520	0.095
Intraoperative hypotension	47 (24)	80 (26)	0.739	0.040	23 (25)	28 (30)	0.511	0.121
Days waiting for operation (day)	3.4 ± 2.7	6.1 ± 39.9	0.227	0.098	3.4 ± 2.7	3.3 ± 2.3	0.813	0.035
Preoperative pan score (NRS)	2.8 ± 0.7	2.8 ± 0.8	0.952	0.005	2.9 ± 0.5	2.9 ± 0.4	0.724	0.053
Postoperative pain score (NRS)	5.5 ± 1.4	5.7 ± 1.5	0.181	0.127	5.3 ± 1.5	5.6 ± 1.5	0.219	0.187

Data are shown as mean ± standard deviation or median (interquartile range) or number (%). NRS: numeric rating scale.

**Table 3 medicina-61-00336-t003:** Incidence of delirium and postoperative data.

	Before Matching				After Matching			
Variables	R Group (n =194)	S Group (n = 308)	*p*	SMD	R Group (n = 93)	S Group (n = 93)	*p*	SMD
Delirium	66 (33.7)	95 (30.7)	0.475	0.068	31 (33.3)	31 (33.3)	1.000	<0.001
Hospital stay	32.5 ± 14.9	38.1 ± 83.2	0.251	0.094	34.6 ± 17.4	32.5 ± 18.5	0.424	0.118
ICU stay	0.13 ± 0.96	0.25 ± 1.5	0.310	0.088	0.03 ± 0.23	0.13 ± 0.96	0.346	0.139
30-day mortality	3 (1.5)	5 (1.6)	1.000	0.006	0 (0)	2 (2.2)	0.497	0.210

Data are shown as mean ± standard deviation or median (interquartile range) or number (%). ICU: intensive care unit.

**Table 4 medicina-61-00336-t004:** Multiple logistic regression analysis for POD incidence.

	Estimate	Standard Error	Wald	*p*	Odds Ratio	95% CI
Remimazolam	0.0755	0.4267	0.177	0.860	1.078	0.467–2.489
Gender (female)	−1.0881	0.4742	−2.295	0.022	0.337	0.133–0.853
Polypharmacy	0.0881	0.0444	1.985	0.047	1.092	1.001–1.191
Hypertension	0.2855	0.6125	0.466	0.641	1.330	0.401–4.419
Hemoglobin	−0.0676	0.1634	−0.414	0.679	0.935	0.679–1.287
Albumin	−1.2099	0.7013	−1.725	0.084	0.298	0.075–1.179
CRP	−0.0171	0.0597	−0.287	0.774	0.983	0.875–1.105

95% CI: 95% confidence interval. The cut-off value of each risk factor from univariate analysis is *p* < 0.1. POD: postoperative delirium.

## Data Availability

The data are presented within the article. Additional data are available on request from the corresponding author.

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
