# Peer review of "The Effects of Remimazolam and Inhalational Anesthetics on the Incidence of Postoperative Hyperactive Delirium in Geriatric Patients Undergoing Hip or Femur Surgery Under General Anesthesia: A Retrospective Observational Study"

_medicina, 2025, doi:10.3390/medicina61020336_

Round 1

Reviewer 1 Report

Comments and Suggestions for Authors

This study explores an important clinical issue, comparing the effects of remimazolam and sevoflurane on postoperative delirium (POD) in elderly patients after hip or femur surgery. Strengths include the use of propensity score matching to ensure reliable group comparisons and a thorough collection of demographic, surgical, and anesthesia-related data. These elements provide a solid foundation for analyzing factors influencing POD, with results that are clear and well-organized.

However, some limitations reduce the study’s impact. The POD assessment relies on observable symptoms and psychiatrist consultations, which may miss hypoactive cases. Preoperative delirium is mentioned as a variable but its role as a confounding factor is unclear. While male gender is identified as a risk factor for POD, the study does not explore possible biological or social explanations. Additionally, confounders like socioeconomic and nutritional status are not addressed, despite efforts to minimize selection bias.

The discussion could benefit from deeper integration with existing research, particularly on how remimazolam compares to other benzodiazepines regarding POD risk. The identification of polypharmacy as a risk factor is useful, but practical strategies for its management could be expanded. The study’s limitations, such as its retrospective design and non-standardized POD assessment methods, should be more critically examined, with recommendations for improvement in future research.

Overall, the study provides valuable insights into remimazolam’s safety and potential benefits for elderly patients. Enhancing the discussion with clearer clinical recommendations, a deeper analysis of risk factors, and stronger connections to existing literature would improve the manuscript’s impact and applicability.

Author Response

Thank you very much for the thoughtful and thorough review. We hope our responses sufficiently answered your concerns. For any points insufficient, please let me know at any time.

Point 1. The POD assessment relies on observable symptoms and psychiatrist consultations, which may miss hypoactive cases. Preoperative delirium is mentioned as a variable but its role as a confounding factor is unclear. While male gender is identified as a risk factor for POD, the study does not explore possible biological or social explanations. Additionally, confounders like socioeconomic and nutritional status are not addressed, despite efforts to minimize selection bias.

Response 1-1. Thank you for your feedback. Our retrospective design may have overlooked the hypoactive type of delirium, that we primarily focused on hyperactive delirium as included in the title. Additionally, while we attempted to adjust for as many confounding variables as possible, the retrospective design of the study made it impossible to assess socioeconomic and nutritional status using the electronic medical records available at our center. Therefore, we have revised the limitation on page 8 as follows:

" While matching did balance the patients’ baseline characteristics, certain unmeasured variables such as socioeconomic and nutritional status could still have impacted the results.”

Response 1-2. Preoperative delirium was included in the logistic regression analysis but was not positively correlated with POD incidence. Since the incidence of preoperative delirium was not significantly different between the groups before and after matching, we did not consider it a confounding factor.

Response 1-3. Regarding gender differences, we have added the following to the discussion on page 8.

"Male gender is a known risk factor for POD following surgery. Previous epidemiological studies suggest that male gender may be exposed to more risk factors affecting cognitive-related brain domains, such as obstructive sleep apnea, alcohol dependence, and psychological stress related to illness, all of which may contribute to POD [20]. Additionally, estrogen may play a protective role in individuals with potential cognitive impairment, although the specific mechanisms underlying its effects on POD remain unclear [21]."

Point 2. The discussion could benefit from deeper integration with existing research, particularly on how remimazolam compares to other benzodiazepines regarding POD risk.

Response 2.  Thank you for your recommendation. There are limitations in studies comparing continuous infusion of remimazolam with other long-acting benzodiazepines during surgery, since there are no benzodiazepine agents suitable for continuous infusion other than remimazolam. Most studies compare remimazolam with inhalational agents or propofol so far. Possible explanations for the differing impact on POD between remimazolam and other benzodiazepines, such as midazolam, is provided in the discussion on page 7. I hope this addresses your comments. Please let me know if any further revisions are needed.

Point 3. The identification of polypharmacy as a risk factor is useful, but practical strategies for its management could be expanded.

Response 3. We strongly agree with your opinion. We found that there are two main tools to help manage polypharmacy for elderly patients undergoing surgery. We described details on page 8 as follows.

“There are tools available to help manage polypharmacy and limit inappropriate medication use in elderly patients. Two commonly used tools are the American Geriatrics Society (AGS) Beers Criteria and the Screening Tool of Older People’s Prescriptions (STOPP) and Screening Tool to Alert to Right Treatment (START), collectively known as the STOPP/START criteria. The AGS Beers Criteria focus on medications that should be used with caution in the elderly and include information on drug-drug interactions. The STOPP/START criteria serve as a guideline for appropriate prescribing to reduce adverse drug events.”

Point 4. The study’s limitations, such as its retrospective design and non-standardized POD assessment methods, should be more critically examined, with recommendations for improvement in future research.

Response 4. Thank you for your comment! We revised the limitation on page 8, and now it reads as follows.

“The second limitation was that POD was identified through observable symptoms and psychiatrist consultations, which may have led to hypoactive POD cases going unreported. Therefore, we are conducting a prospective study using standardized POD assessment tools, such as Confusion Assessment Method and other supplements, in collaboration with neuropsychiatrists to ensure that all subtypes are detected.”

Reviewer 2 Report

Comments and Suggestions for Authors

This study suggests that remimazolam is no more likely to cause postoperative delirium (POD) in geriatric patients undergoing hip surgery than sevoflurane. Despite concerns about benzodiazepines increasing POD risk, emerging evidence indicates that remimazolam is a safe alternative for elderly surgical candidates.

Key Findings:

  • Previous studies (e.g., Fujimoto et al., 2024, and Yang et al., 2023) showed that remimazolam results in fewer POD incidences than propofol, sevoflurane, and desflurane in various surgeries.
  • Unique characteristics of remimazolam, such as its ultra-short half-life and rapid elimination, may explain its lower POD risk. It also exhibits inflammation-attenuating effects and hemodynamic stability, which contribute to better postoperative outcomes.
  • Elderly patients benefit from remimazolam's reduced hypotension risk during anesthesia induction, maintaining stable cerebral blood flow and minimizing POD risk.

Risk Factors and Recommendations:

  • Polypharmacy and male gender were identified as risk factors for POD, highlighting the importance of minimizing unnecessary medications in elderly patients during the perioperative period.
  • No correlation was found between Charlson's comorbidity index and POD incidence, suggesting drug interactions as a primary factor.

Limitations:

  • The study’s retrospective design may have introduced selection bias, despite propensity score matching.
  • Reliance on observable symptoms and psychiatrist consultations for POD diagnosis could have missed hypoactive cases. Future prospective studies with standardized tools are recommended.

Conclusion:
Remimazolam demonstrates significant advantages for elderly patients, particularly in high-risk surgeries, due to its safety profile, hemodynamic stability, and potential to reduce POD incidence.

However:

This manuscript examines anesthetic strategies for hip replacement surgery, emphasizing their implications for elderly patients. While the presented data aligns with existing literature in respect to GA, the study contributes minimally to advancing knowledge.

Missing Key Findings:

  1. Regional anesthesia (RA) demonstrated benefits, including reduced complications and faster recovery in elderly patients.
  2. Combining RA with general anesthesia (GA) yielded better outcomes than GA alone, particularly in terms of postoperative pain management and mobility restoration.

My Discussion Recommendations: The manuscript could benefit from an expanded discussion on:

  • Comparing alternative anesthetic procedures, such as RA alone versus GA alone or RA combined with GA.
  • Addressing the limitations of GA in elderly populations and the synergistic advantages of RA-GA combinations.
  • Contextualizing findings within the broader literature to enhance the manuscript's contribution to clinical practice.

Conclusion: While the study provides valuable insights into anesthetic options, its publishability hinges on a deeper comparative analysis of methods and integration into the existing body of evidence.

Comments on the Quality of English Language

because I am not a native speaker, I suggest the editor and the Journal to check for misspelling and wording....

Author Response

Thank you very much for the thoughtful and thorough review. We hope our responses sufficiently answered your concerns. For any points insufficient, please let me know at any time.

Point 1. The study’s retrospective design may have introduced selection bias, despite propensity score matching. Reliance on observable symptoms and psychiatrist consultations for POD diagnosis could have missed hypoactive cases. Future prospective studies with standardized tools are recommended.

Response 1. Thank you for your comments. Our retrospective design may have overlooked the hypoactive type of delirium. We have acknowledged this as a limitation that we primarily focused on hyperactive delirium as included in the title. Additionally, while we attempted to adjust for as many confounding variables as possible, the retrospective design of the study made it impossible. Also, we are conducting a prospective study with standardized POD assessment tools with neuropsychiatrists. We have revised the limitation on page 8-9 as follows.

" While matching did balance the patients’ baseline characteristics, certain unmeasured variables such as socioeconomic and nutritional status could still have impacted the results. Hence, as a retrospective single-center study, caution is needed when generalizing the findings more broadly. The second limitation was that POD was identified through observable symptoms and psychiatrist consultations, which may have led to hypoactive POD cases going unreported. Therefore, we are conducting a prospective study using standardized POD assessment tools, such as Confusion Assessment Method and other supplements, in collaboration with neuropsychiatrists to ensure that all subtypes are detected.”

Point 2. The manuscript could benefit from an expanded discussion on: Comparing alternative anesthetic procedures, such as RA alone versus GA alone or RA combined with GA. Addressing the limitations of GA in elderly populations and the synergistic advantages of RA-GA combinations. Contextualizing findings within the broader literature to enhance the manuscript's contribution to clinical practice.

Response 2. We fully agree with your comments. Peripheral nerve block is undoubtedly important in preventing POD. However, we primarily excluded patients who received a peripheral nerve block to focus on the impact of remimazolam on POD, which is why we initially did not include nerve blocks in the discussion. We have now addressed the role of peripheral nerve blocks combined with general anesthesia, as well as the potential benefits of combining peripheral nerve blocks with remimazolam, which we anticipate will result in the lowest incidence of POD. As per your recommendation, we believe adding this paragraph enhances the manuscript’s contribution in alignment with current clinical practice. Now it reads as follows and is added on page 8.

“Peripheral nerve block is undoubtedly important in preventing POD, although we excluded patients who received a peripheral nerve block to focus on the impact of remimazolam on POD. There is consistent evidence that combining a peripheral nerve block with general anesthesia reduces the incidence of POD in elderly patients with hip fractures [24, 25]. A meta-analysis of 19 RCTs demonstrated that peripheral nerve blocks significantly lowered POD incidence (OR: 0.59, 95% CI [0.40–0.87], p = 0.007) [26]. Specifically, the fascia iliaca compartment block (OR: 0.58, 95% CI [0.37–0.91], p = 0.02) and the femoral nerve block (OR: 0.33, 95% CI [0.11–0.99], p = 0.05) were associated with reduced POD risk. Additionally, peripheral nerve blocks significantly reduced pain scores (p = 0.002). Pain is a major risk factor for POD, and by utilizing peripheral nerve blocks, elderly patients benefit from better analgesia, reduced inflammatory response, decreased opioid consumption, and ultimately, a lower incidence of POD. Without peripheral nerve block, postoperative pain management after general anesthesia is challenging. In future studies, we may explore the combination of peripheral nerve blocks with remimazolam, which we anticipate will result in the lowest POD incidence. If confirmed, this approach could be widely implemented in clinical practice for elderly patients.”

Round 2

Reviewer 2 Report

Comments and Suggestions for Authors

The revisions in the manuscript are acceptable, and the authors have thoughtfully incorporated my comments.